# Identification of *Somatostatin Receptor Subtype 1 (SSTR1)* Gene Polymorphism and Their Association with Growth Traits in Hulun Buir Sheep

**DOI:** 10.3390/genes13010077

**Published:** 2021-12-28

**Authors:** Xue Li, Ning Ding, Zhichao Zhang, Dehong Tian, Buying Han, Sijia Liu, Dehui Liu, Fei Tian, Kai Zhao

**Affiliations:** 1Key Laboratory of Adaptation and Evolution of Plateau Biota, Northwest Institute of Plateau Biology, Chinese Academy of Sciences, Xining 810001, China; lixue@nwipb.cas.cn (X.L.); dingning@nwipb.cas.cn (N.D.); zczhang@genetics.ac.cn (Z.Z.); tiandehong@nwipb.cas.cn (D.T.); hanbuying@nwipb.cas.cn (B.H.); liusj@nwipb.cas.cn (S.L.); liudehui@nwipb.cas.cn (D.L.); tianfei@nwipb.cas.cn (F.T.); 2University of Chinese Academy of Sciences, Beijing 100049, China; 3Qinghai Provincial Key Laboratory of Animal Ecological Genomics, Northwest Institute of Plateau Biology, Chinese Academy of Sciences, Xining 810008, China

**Keywords:** *SSTR1*, association, growth traits, Hulun Buir sheep

## Abstract

This study was conducted to evaluate *SSTR1* gene polymorphisms and their association with growth traits in Hulun Buir sheep. We followed 233 Hulun Buir sheep from birth to 16 months of age, born in the same pasture and on the same year under a consistent grazing conditions. The body weight (BW), body height (BH), body length (BL), chest circumference (ChC), chest depth (ChD), chest width (ChW), hip width (HW), and cannon circumference (CaC) were measured and recorded at birth, 4 months, 9 months, and 16 months of age. The polymorphisms of the *SSTR1* gene in Hulun Buir sheep were excavated using exon sequencing, and association analyses of between SNPs and growth traits at each growth stage were conducted. The results showed that there were four SNPs in Exon 2 of the *SSTR1* gene, SNP1, SNP2, and SNP3 were low mutation sites, and SNP4 was a moderate mutation site. Four SNPs were consistent with Hardy–Weinberg equilibrium, and all of them were synonymous mutations. The association analyses found that the genotypes of SNP2 were significantly associated with WW and BH at 4 months of age, BW, BL, ChC, and HW at 9 months of age (*p* < 0.05), and extremely significantly associated with ChD at 4 and 9 months of age (*p* < 0.01). There were significant associations between SNP3 and BH at 9 months of age, between SNP4 and ChD, ChW, and CaC at 9 months of age, and BW and ChC at 16 months of age (*p* < 0.05). There were no detectable associations with growth traits among the seven haplotypes between the SNP1, 3, and 4 of a strong linkage disequilibrium (*p* > 0.05). These results indicated that SNP2, SNP3, and SNP4 may be used as molecular markers for growth traits of Hulun Buir sheep.

## 1. Introduction

The Hulun Buir sheep is a famous native breed in China, and is distributed in Hulun Buir city, Inner Mongolia. It has the advantages of outstanding stress resistance, strong adaptability, and a particularly high quality of meat with low fat and a variety of amino acids. As a traditional mutton sheep breed, Hulun Buir sheep has not been selected using advanced breeding methods, thus showing low productivity, a slow growth rate, and a low slaughter rate. Many candidate genes have been reported to regulate metabolism and control the growth rate of domestic animals [1,2,3]. Single-nucleotide variant in these genes were widely used as molecular markers, which accelerated the breeding process and improved productivity.

Somatostatin (SST), also known as growth hormone-inhibiting hormone or somatotropin release-inhibiting factor, is considered as a hypothalamic factor that inhibits the secretion of growth hormone (GH) [4]. In mammals, there are five somatostatin receptor subtypes (SSTR1-5) [5,6]; SST and SSTRs are widely distributed in the central nervous system, pancreas, intestines, stomach, kidney, liver, pancreas, lungs, and placenta, and are involved in diverse biological effects [7], but it only works when it binds to G-protein-coupled SSTRs [8]. The mechanisms by which SSTRs regulate growth and developments are complex and coordinated by many mechanisms [9]. Firstly, SSTRs represses the secretion of prolactin, thyroid-stimulating hormone (TSH) [10,11], the stomach secrete hormone, growth hormone releasing hormone (GHRH), secretin, glucagon, insulin, and SST itself in the pancreas [12]. Secondly, SSTRs decrease the nutrient absorption rate in the gastrointestinal tract by inhibiting the secretion of gastrointestinal hormones and digestive enzymes [13]. Besides, they control digestion and absorption rates by reducing gastrointestinal motility, gallbladder contraction, and blood flow, which also in turn affect feed conversion and growth characteristics [14].

The nucleotide sequence of the *SSTR1* gene is highly conserved among species, and ovine *SSTR1* shares 88% sequence homology with human, 84% homology with rat, and 87% homology with mouse [15]. *SSTR1* plays a more important role in insulin and GH secretion than other receptors, particularly in maintaining basal levels of GH [16]. It has been reported that suppressed *SSTR1* expression leads to weight loss and a decreased growth rate in mice [17,18]. In humans, *SSTR1* has been revealed to be associated with cancers and tumors, and its selective agonists have been used as cancer therapies [19]. In domestic animals, genetic variants have been identified in the *SSTR1* gene of goat and the 3’UTR region of New Zealand Romney sheep were associated with growth traits [20,21].

Considering the importance of *SSTR1* in controlling the growth hormone axis, and the lack of research on the effect of the *SSTR1* gene on growth performance in sheep, the genetic polymorphisms in this gene of Hulun Buir sheep were examined, the association between each genotype and its growth traits were analyzed in this study.

## 2. Materials and Methods

### 2.1. Sample Collection and DNA Isolation

The experimental population consisted of 233 purebred Hulun Buir sheep (124 females and 109 males), which were raised on one farm under the same feeding and management conditions of grazing. The growth traits of each sheep were recorded from birth to adulthood at 16 months of age, including birth weight (BRW), body weight (BW), body length (BL), body height (BH), chest circumference (ChC), chest depth (ChD), chest width (ChW), hip width (HW), and cannon circumference (CaC). All animal experiments were conducted following the procedures described in the “Guidelines for animal care and use” manual, and were approved by the Animal Care and Use Committee, Northwest Institute of Plateau Biology, Chinese Academy of Sciences.

### 2.2. Primer Design and Sequencing

Ear tissue was collected and preserved in 75% alcohol. DNA was purified using A DNA extraction kit (TIANGEN, Beijing, China). Primers were designed for all exons of *SSTR1* gene using Primer3 v0.4.0(1) [22]. The *SSTR1* gene transcript has three exons (ENSOART00020017811.1), and one primer pair was designed covering exons 1 to 2 and another one between exons 2 and 3. Information on the primers is presented in Table 1.

PCR amplifications were carried out in a 30-μL reaction consisting of 100 ng of DNA, 15 μL of 2 × Taq PCR Master Mix, 1 μL of each primer, and double-distilled water (dH_2_O) to make up the volume. Amplifications were performed using Bio-rad S1000 thermal cyclers (Bio-Rad, Hercules, CA, USA). The thermal profile was as follows: initial denaturation at 94 °C for 2 min, followed by 35 cycles 94 °C for 10 s (denaturation), 60 °C for 30 s (annealing), 72 °C for 60 s (elongation), with a final extension step at 72 °C for 5 min. The PCR products were visualized using 1.0% agarose gel electrophoresis to determine amplicon quality and quantity. The sequencing was performed to identify mutations using Sanger sequencing (Applied Biosystems, Foster City, CA, USA). DNAMAN (version 5.2.10, Lynnon Biosoft, Montreal, QC, Canada) was used to conduct sequence analyses.

### 2.3. Population Genetic Analyses

Population genetic indexes, including allele frequency, heterozygosity (He), observed heterozygosity (Ho), effective allele numbers (Ne), and the polymorphism information content (PIC) were analyzed according to Nei’s methods [23]. Hardy–Weinberg equilibrium (HWE) was tested for genotypes of SNPs using web-based software (http://scienceprimer.com/hardy-weinberg-equilibrium-calculator, accessed on 20 November 2021). Linkage didequilibrium (LD) analysis and haplotypes were assessed using HAPLOVIEW (v.4.2) [24].

### 2.4. Statistical Analysis

SPSS Statistics (version 19, IBM, Armonk, NY, USA) was used to perform all analyses and values were expressed as mean ±  SE (standard error). General linear mixed models (GLMMs) were carried out to examine the associations between the genotypes and individual growth traits, and statistical significance was defined at *p* < 0.05. In this model, genotype and gender are fixed factors, the interaction between these two fixed factors was judged by the test of the intersubjective effect. If there is an interaction between genotype and gender, the follow statistical model was used:Y = μ + Genotype + Gender + Combination + ε
where:Y is the trait measured on each animal (BW, BL, BH, ChW, ChD, ChW, HW, and CaC);μ is the mean for the trait;Genotype is the genotype effect;Gender is the gender effect;Combination is the combination effect of the gender and genotype;ε is the random error and is assumed to be independent, N (0, σ2) distribution.

If there is no interaction between genotype and gender, the following statistical model was used:Y = μ + Genotype + ε
where:Y is the trait measured on each animal (BW, BL, BH, ChW, ChD, ChW, HW, and CaC);μ is the mean for the trait;Genotype is the genotype effect;ε is the random error and is assumed to be independent, N (0, σ2) distribution.

## 3. Results

### 3.1. Polymorphism in SSTR1 Gene

A total of four SNPs were identified in the *SSTR1* gene of Hulun Buir sheep have been deposited in dbSNP database, which were defined as SNP1 (A345G, rs415509729), SNP2 (A285G, rs426187704), SNP3 (C309T, rs404696179), and SNP4 (G951C, rs405457403). These SNPs were synonymous mutations located on exon 2 of *SSTR1*. The sequenced peak maps for four SNPs are illustrated in Figure 1, Figure 2, Figure 3 and Figure 4.

### 3.2. Population Genetics and the Linkage Disequilibrium Analysis

Ne was calculated for each SNP, ranging from 1 to 2. The allele frequency of SNPs met the Hardy–Weinberg equilibrium balance (*p* > 0.05). The PIC indicated that SNP4 was a moderate polymorphism, and SNP1–3 were classified as having a low polymorphic locus (Table 2). Linkage disequilibrium analysis showed a strong linkage disequilibrium (D’ > 0.85) between SNP1, SNP3, and SNP4 (Figure 5), and a total of seven common seven haplotypes were identified in this linkage disequilibrium region.

### 3.3. Association Analysis of Genetic Variants and Haplotypes in SSTR1 with Growth Traits of Hulun Buir Sheep

The effect of genotypes on phenotypic value was considered in this study, and association studies were conducted in 233 sheep to assess the effects of SNPs on the growth traits at birth and at 4 months of age (Table 3), 9 months of age (Table 4), and 16 months of age (Table 5) of age. Table 6, Table 7 and Table 8 show the results of association analyses between the haplotypes and growth traits at birth and 4, 9, and 16 months of age, respectively.

#### 3.3.1. Association Analysis of SSTR1 with Growth Traits at Birth and 4 Months of Age in Hulun Buir Sheep

The analyses found that the genotypes of SNP2 were significantly associated with WW and BH (*p* < 0.05) and extremely significantly associated with ChD at 4 months of age (*p* < 0.01). SNP1, SNP3, and SNP4 were not significantly correlated with all the phenotypes (*p* > 0.05; Table 3).

#### 3.3.2. Association Analysis of SSTR1 with Growth Traits at 9 Months in Hulun Buir Sheep

Among the phenotypes at 9 months, we identified a significant association between SNP2 and BW, BL, ChC, and HW, between SNP3 and BH, as well as between SNP4 and ChD, ChW, and CaC (*p* < 0.05). Additionally, SNP2 was extremely significantly associated with ChD (*p* < 0.01; Table 4).

#### 3.3.3. Association Analysis of SSTR1 with Growth Traits at 16 Months in Hulun Buir Sheep

Among the phenotypes at 16 months, SNP3 had a significant association with BW and ChC (*p* < 0.05), and had a highly significant association with BL and BH (*p* < 0.01). No significant differences were observed between the rest of the SNPs with other phenotypic data (*p* > 0.05; Table 5).

#### 3.3.4. Haplotype Association Analysis with Growth Traits

Although we discovered seven highly linked haplotypes among the four SNPs, none showed a significant association with any phenotypes from birth to 16 months old (*p* > 0.05; Table 6, Table 7 and Table 8).

## 4. Discussion

Growth is one of the most important economical traits monitored in domestic animals, and therefore is the main objective of most genetic-selection programs [25]. As the main measure of growth traits, body weight and body measurements have important impacts on the production of meat and wool [26]. In this study, we recorded these productive traits from birth to adulthood in Hulun Buir sheep, which included body weight and seven body size indicators at 4, 9, and 16 months. The phenotypic data contained the main growth traits of the sheep development, which provided a comprehensive understanding of growth trends in Hulun Buir sheep. Therefore, because our data were so comprehensive, and the experimental time so long, there are very few other studies that are comparable.

Single nucleotide polymorphisms (SNPs), defined as a substitution, insertion, or deletion of a single nucleotide, are important genetic sources for animal breeding, which regulate gene expression and protein functions, depending on the location of the SNP in regulatory sequences or coding regions [27]. Exons are protein-coding regions consisting of only 1–2% of the genome, and the mutation rate in exons is approximately 1/5 that of non-coding regions [28]; however, almost 85% of reported disease-causing genes harbored mutations in their exons, which is of great significance in the study of genetic diseases [29]. Exome sequencing, thus, is the most efficient approach to identify potentially functional mutations, which might be related to phenotypes in domestic animals [30]. Thus, exome sequencing is the most cost-efficient sequencing approaches for conducting genome research and animal phenotyping [31]. According to the above arguments, we used exon sequencing technology to identify SNPs associated with growth traits more effectively.

Linked loci are of particular concern as there is substantial linkage disequilibrium between causal SNPs [32]. Studies have shown that body size is maintained by the build-up of inter-population linkage disequilibrium between loci, caused by selection [32]. In our study, there was a linkage disequilibrium between SNP1, SNP3, and SNP4, seven haplotypes were formed from three linked SNPs, but there was no association between growth traits. This result may be due to the small sample size, or the SNPS in this gene may interact with other genes.

To discover the potential functional mutations related to the growth traits in Hulun Buir sheep, we conducted exon sequencing of the *SSTR1* gene, The ovine *SSTR1* gene transcript has three exons and two introns (ENSOART00020017811.1) located on chromosome 18 (GenBank, Gene ID: 443202), encoding 1089 bp base (47398442–47400840) and 362 amino acid residues [33]. As a receptor of somatostatin, the *SSTR1* gene plays an important role in many physiological processes, such as cell anti-proliferation, and inhibition of gastrointestinal motility and regulates a variety of signal transduction pathways [34]. Four SNPs (A345G, A285G, C309T and G951C) were all in Hardy–Weinberg equilibrium, which indicated an adequate population size of Hulun Buir sheep under random mating (or without selection) in the experiment [35]. None of them caused a change in amino acid sequence, known as synonymous SNPs [36]. Several lines of evidence suggested that synonymous mutations can affect translation kinetics and protein folding, which lead to phenotypic changes [37,38]. Although all these SNPs in *SSTR1* were synonymous mutations, our analyses suggested these SNPs had a significant association with the growth traits of Hulun Buir sheep at 4, 9, and 16 months of age.

*SSTR1* is an important peptide hormone that regulates diverse functions, including cell proliferation, neurotransmission, and particularly, inhibiting the release of growth hormone [39]. Therefore, *SSTR1* is a potential candidate gene for growth traits in livestock. Jin et al. identified the sequence variations in the PS2 locus of caprine *SSTR1*, which were associated with growth traits, including BL, BH, CaC, and, ChC [20]. The SNPs in the 3′-UTR of ovine *SSTR1* were associated with BW, hot carcass weight, and VIAscan fat depth at the 12th rib [21]. In the current study, we discovered new mutations in the coding region of *SSTR1* in Hulun Buir sheep, indicating the potential application of the *SSTR1* gene in the sheep breeding.

In our results, SNP2 was found to be significantly correlated with WW, BH, and ChD at 4 months of age, and BW, BL, ChC, ChD, and HW at 9 months. SNP4 was significantly correlated with ChW, ChD, and CaC at 9 months. SNP3 was significantly associated with BW, BL, BH, and ChC at 16 months of age. These results suggest that SNP2 and SNP4 may mainly affect the growth traits of sheep in the early growth period (4 to 9 months of age), while SNP3 may affect the growth traits in the mature period (16 months of age). Therefore, we can infer that SNP2 and SNP4 may be used as molecular markers for early growth and development traits of Hulun Buir sheep, and SNP3 may be used for the late growth period. Although we found seven haplotypes from three linked SNPs, none of them were associated with growth traits in the experimental population. Further analyses, including increasing the population size and identifying more SNPs, are required to explore the associations between haplotypes and growth traits in Hulun Buir sheep.

## 5. Conclusions

In the current study, we applied exon sequencing technology to screen the *SSTR1* gene and discovered four SNPs in Hulun Buir sheep for the first time. Despite synonymous mutations in all SNPs, SNP2, SNP3, and SNP4 were associated with the BW, BH, BL, ChC, ChW, HW, and CaC in 4-, 9-, and 16-month-old Hulun Buir sheep, which indicated that these SNPs could be used as molecular markers for the selection of growth traits in Hulun Buir sheep. Our analyses also suggested that local Chinese sheep breeds harbor important genetic resources, which have potential implications in animal breeding.

## Figures and Tables

**Figure 1 genes-13-00077-f001:**
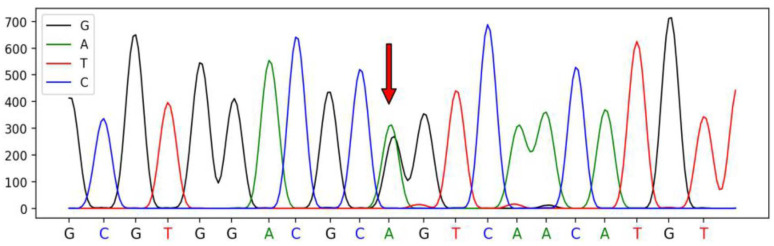
The sequencing peak map of the *SSTR1* and the mutated SNP1 site of Hulun Buir sheep. The sites marked by the red arrow was the SNP1 mutation, which was found and identified in exon 2 of *SSTR1*, A345G (rs415509729). The sequences were analyzed using DNAMAN software.

**Figure 2 genes-13-00077-f002:**
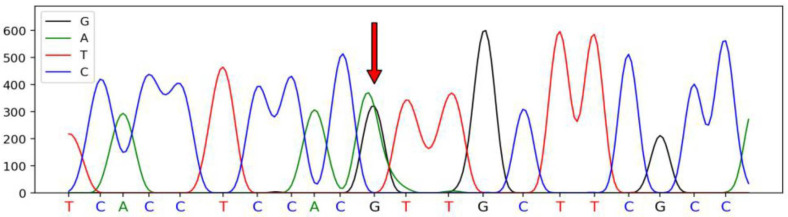
The sequencing peak map of the *SSTR1* and the mutated SNP2 site of Hulun Buir sheep. The sequencing peak map of the *SSTR1* and the mutated SNP2 site of Hulun Buir sheep. The sites marked by the red arrow was the SNP2 mutation, which was found and identified in exon 2 of *SSTR1*, A285G (rs426187704). The sequences were analyzed using DNAMAN software.

**Figure 3 genes-13-00077-f003:**
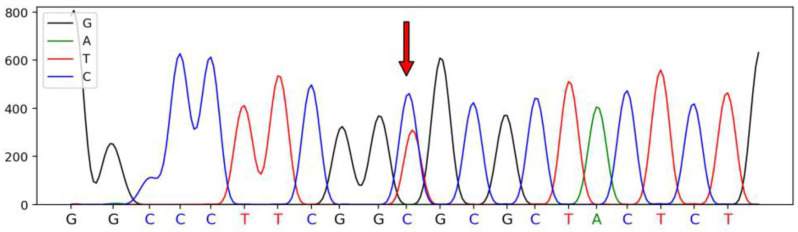
The sequencing peak map of the *SSTR1* and the mutated SNP3 site of Hulun Buir sheep. The sites marked by the red arrow was the SNP3 mutation, which was found and identified in exon 2 of *SSTR**1*, C309T (rs404696179). The sequences were analyzed using DNAMAN software.

**Figure 4 genes-13-00077-f004:**
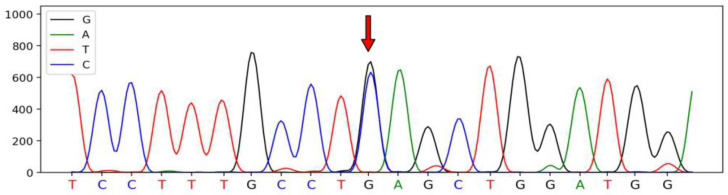
The sequencing peak map of the *SSTR1* and the mutated SNP4 site of Hulun Buir sheep. The sites marked by the red arrow was the SNP4 mutation, which was found and identified in exon 2 of *SSTR**1*, G951C (rs405457403). The sequences were analyzed using DNAMAN software.

**Figure 5 genes-13-00077-f005:**
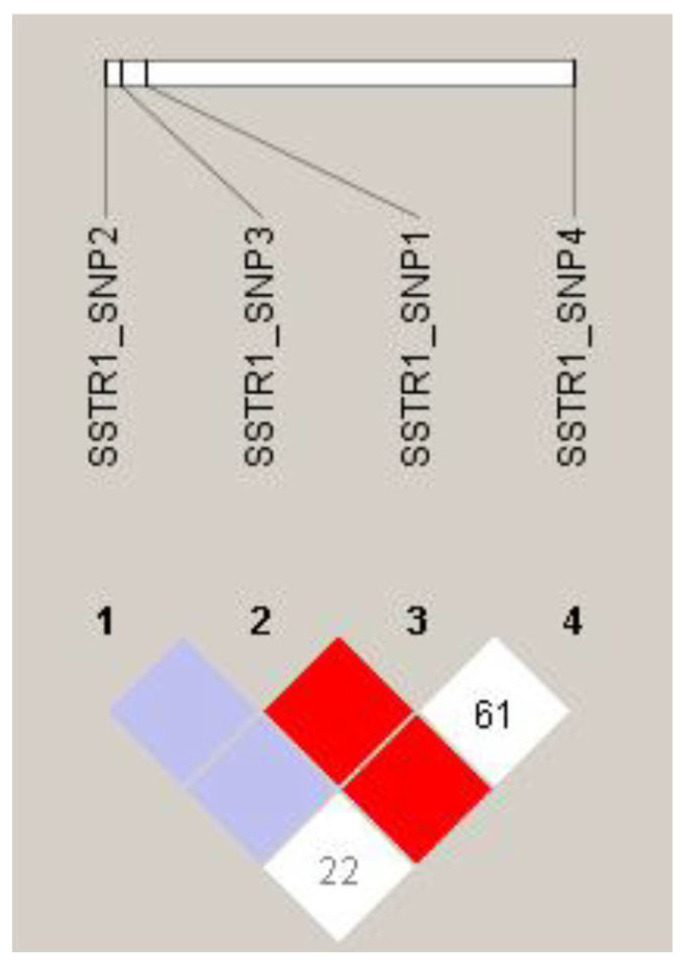
The linkage disequilibrium plot of SNPs in *SSTR**1* of in Hulun Buir sheep. Linkage disequilibrium (LD) estimated among *SSTR1* variations in Hulun Buir sheep. The R2 values indicated the group of SNP1, SNP3, and SNP4.

**Table 1 genes-13-00077-t001:** Primer information of *SSTR1* of Hulun Buir sheep.

Primer Names	Primer Sequences (5′–3′)	Size (bp)	Tm (°C)
E1-2	F:ACATCCTCAACCTGGCCATC	589	60
	R:AGCTGCACCACATAGAAGGG		
E1-3	F:CATGGGCTTCCTGCTGC	558	60

**Table 2 genes-13-00077-t002:** Population genetics analyses of *SSTR1* of in Hulun Buir sheep ^1^.

SNP	Allele Frequency			Ho	He	PIC	Ne	HWE
	AA	AB	BB					
SNP1(A/G)	0.8233	0.1767	0	0.1612	0.1767	0.1482	1.2146	0.2988
SNP2(A/G)	0.9485	0.0515	0	0.0472	0.0503	0.0490	1.0530	0.1198
SNP3(C/T)	0.8534	0.1466	0	0.1466	0.1359	0.1266	1.1573	0.2621
SNP4(G/C)	0.5193	0.4034	0.0773	0.4034	0.4023	0.3207	1.6731	0.5920

He = Heterozygosity; Ho = observed heterozygosity; PIC = polymorphism information content; Ne = effective allele numbers; HWE = Hardy–Weinberg equilibrium. ^1^ A group size of Population genetics analyses was *n* = 233.

**Table 3 genes-13-00077-t003:** Association analyses of SNPs and genotypes in *SSTR1* with growth traits of Hulun Buir sheep at birth and weaning ^1^.

SNP	Genotype Frequency	BRW/kg	BW/kg	BL/cm	BH/cm	ChC/cm	ChD/cm	ChW/cm	HW/cm	CaC/cm
SNP1	AA (*n* = 191)	4.20 ± 0.05	23.47 ± 0.51	57.26 ± 0.50	55.98 ± 0.39	68.44 ± 0.54	28.07 ± 0.22	15.83 ± 0.17	12.48 ± 0.11	7.47 ± 0.04
	AG (*n* = 41)	4.24 ± 0.11	22.99 ± 1.11	56.95 ± 1.10	56.17 ± 0.86	68.35 ± 1.18	28.24 ± 0.48	15.54 ± 0.36	12.31 ± 0.23	7.49 ± 0.09
	GG (*n* = 0)	/	/	/	/	/	/	/	/	/
SNP2	AA (*n* = 221)	4.21 ± 0.05	23.65 ± 0.47 ^a^	57.39 ± 0.47	56.21 ± 0.36 ^a^	68.64 ± 0.50	28.24 ± 0.20 ^A^	15.74 ± 0.15	12.42 ± 0.10	7.49 ± 0.04
	AG (*n* = 12)	4.18 ± 0.20	19.28 ± 2.03 ^b^	53.87 ± 2.01	52.81 ± 1.56 ^b^	65.31 ± 2.16	25.88 ± 0.87 ^B^	16.68 ± 0.67	13.13 ± 0.43	7.27 ± 0.17
	GG (*n* = 0)	/	/	/	/	/	/	/	/	/
SNP3	CC (*n* = 198)	4.23 ± 0.05	23.68 ± 0.50	57.41 ± 0.49	56.20 ± 0.38	68.79 ± 0.53	28.18 ± 0.21	15.91 ± 0.16	12.52 ± 0.10	7.49 ± 0.04
	CT (*n* = 34)	4.10 ± 0.12	21.96 ± 1.20	55.86 ± 1.19	55.16 ± 0.92	66.68 ± 1.27	27.70 ± 0.52	15.17 ± 0.39	12.07 ± 0.25	7.41 ± 0.10
	TT (*n* = 0)	/	/	/	/	/	/	/	/	/
SNP4	CC (*n* = 121)	4.16 ± 0.06	22.93 ± 0.64	56.56 ± 0.63	55.60 ± 0.49	67.86 ± 0.67	27.94 ± 0.27	15.66 ± 0.21	12.45 ± 0.13	7.44 ± 0.05
	CG (*n* = 94)	4.27 ± 0.07	23.91 ± 0.73	57.81 ± 0.72	56.66 ± 0.56	69.00 ± 0.77	28.20 ± 0.31	15.99 ± 0.24	12.46 ± 0.15	7.53 ± 0.06
	GG (*n* = 18)	4.18 ± 0.16	24.06 ± 1.66	58.39 ± 1.63	55.73 ± 1.27	69.72 ± 1.75	28.89 ± 0.71	15.69 ± 0.54	12.46 ± 0.35	7.41 ± 0.14

BRW = birth weight; BW = body weight; BL = body length; BH = body height; ChC = chest circumference; ChD = chest depth; ChW = chest width, HW = hip width; CaC = cannon circumference. ^a,b^ Within a row, means with different superscript letters are significantly different (*p* < 0.05). ^A,B^ Within a row, means with different superscript letters are very significantly different (*p* < 0.01). ^1^ Data represent means ± SEM (*n* = 233).

**Table 4 genes-13-00077-t004:** Association analyses of SNPs and genotypes in *SSTR1* with growth traits of Hulun Buir sheep at 9 months of age ^1^.

SNP	Genotype Frequency	BW/kg	BL/cm	BH/cm	ChC/cm	ChD/cm	ChW/cm	HW/cm	CaC/cm
SNP1	AA (*n* = 191)	32.33 ± 0.55	66.51 ± 0.37	63.59 ± 0.31	83.11 ± 0.57	32.64 ± 0.22	21.55 ± 0.19	14.69 ± 0.12	7.56 ± 0.04
	AG (*n* = 41)	31.77 ± 1.24	67.32 ± 0.84	64.32 ± 0.70	84.07 ± 1.29	32.55 ± 0.49	21.99 ± 0.42	14.42 ± 0.26	7.59 ± 0.10
	GG (*n* = 0)	/	/	/	/	/	/	/	/
SNP2	AA (*n* = 221)	32.52 ± 0.51 ^a^	66.83 ± 0.35 ^a^	63.84 ± 0.29	83.52 ± 0.53 ^a^	32.81 ± 0.20 ^A^	21.58 ± 0.18	14.70 ± 0.11 ^a^	7.58 ± 0.04
	AG (*n* = 12)	27.84 ± 2.18 ^b^	63.84 ± 1.47 ^b^	61.68 ± 1.23	78.90 ± 2.26 ^b^	29.70 ± 0.85 ^B^	22.46 ± 0.75	13.74 ± 0.46 ^b^	7.31 ± 0.17
	GG (*n* = 0)	/	/	/	/	/	/	/	/
SNP3	CC (*n* = 198)	32.50 ± 0.54	66.87 ± 0.37	63.96 ± 0.30 ^a^	83.44 ± 0.57	32.74 ± 0.22	21.72 ± 0.19	14.67 ± 0.11	7.59 ± 0.04
	CT (*n* = 34)	31.07 ± 1.29	65.64 ± 0.87	62.41 ± 0.72 ^b^	82.39 ± 1.36	32.07 ± 0.52	21.17 ± 0.44	14.50 ± 0.27	7.48 ± 0.10
	TT (*n* = 0)	/	/	/	/	/	/	/	/
SNP4	CC (*n* = 121)	31.87 ± 0.69	66.43 ± 0.47	63.07 ± 0.38	82.57 ± 0.72	32.34 ± 0.27 ^b^	21.34 ± 0.23 ^b^	14.56 ± 0.15	7.57 ± 0.05 ^b^
	CG (*n* = 94)	32.59 ± 0.80	66.84 ± 0.54	64.33 ± 0.45	83.92 ± 0.83	32.76 ± 0.32 ^b^	21.77 ± 0.27 ^b^	14.73 ± 0.17	7.49 ± 0.06 ^b^
	GG (*n* = 18)	33.03 ± 1.78	67.36 ± 1.20	65.03 ± 0.99	84.65 ± 1.84	34.03 ± 0.70 ^a^	22.88 ± 0.60 ^a^	14.78 ± 0.37	7.94 ± 0.14 ^a^

BW = body weight; BL = body length; BH = body height; ChC = chest circumference; ChD = chest depth; ChW = chest width, HW = hip width; CaC = cannon circumference. ^a,b^ Within a row, means with different superscript letters are significantly different (*p* < 0.05). ^A,B^ Within a row, means with different superscript letters are very significantly different (*p* < 0.01). ^1^ Data represent means ± SEM (*n* = 233).

**Table 5 genes-13-00077-t005:** Association analyses of SNPs and genotypes in *SSTR1* with growth traits of Hulun Buir sheep at 16 months of age ^1^.

SNP	Genotype Frequency	BW/kg	BL/cm	BH/cm	ChC/cm	ChD/cm	ChW/cm	HW/cm	CaC/cm
SNP1	AA (*n* = 191)	38.01 ± 0.48	72.61 ± 0.53	67.35 ± 0.37	81.78 ± 0.40	33.60 ± 0.34	19.44 ± 0.25	18.01 ± 0.16	8.18 ± 0.03
	AG (*n* = 41)	39.19 ± 1.03	73.87 ± 1.14	67.22 ± 0.79	82.36 ± 0.85	34.10 ± 0.42	19.25 ± 0.50	18.42 ± 0.33	8.13 ± 0.06
	GG (*n* = 0)	/	/	/	/	/	/	/	/
SNP2	AA (*n* = 221)	38.48 ± 0.45	72.83 ± 0.50	67.42 ± 0.34	82.06 ± 0.37	32.77 ± 0.18	19.41 ± 0.22	18.13 ± 0.15	8.19 ± 0.03
	AG (*n* = 12)	35.00 ± 1.81	72.85 ± 2.00	66.11 ± 1.38	79.57 ± 1.49	32.48 ± 0.74	19.07 ± 0.87	17.43 ± 0.59	7.99 ± 0.11
	GG (*n* = 0)	/	/	/	/	/	/	/	/
SNP3	CC (*n* = 198)	38.63 ± 0.47 ^a^	73.32 ± 0.51 ^A^	67.81 ± 0.35 ^A^	82.23 ± 0.38 ^a^	33.80 ± 0.19	19.44 ± 0.22	18.14 ± 0.15	8.18 ± 0.03
	CT (*n* = 34)	35.94 ± 1.21 ^b^	69.77 ± 1.32 ^B^	64.39 ± 0.90 ^B^	79.85 ± 1.00 ^b^	33.00 ± 0.50	19.14 ± 0.58	17.79 ± 0.39	8.14 ± 0.07
	TT (*n* = 0)	/	/	/	/	/	/	/	/
SNP4	CC (*n* = 121)	38.36 ± 0.60	72.35 ± 0.66	67.06 ± 0.46	81.73 ± 0.50	33.71 ± 0.24	19.34 ± 0.29	18.08 ± 0.19	8.18 ± 0.04
	CG (*n* = 94)	38.13 ± 0.70	73.66 ± 0.76	67.69 ± 0.53	82.13 ± 0.57	33.83 ± 0.28	19.57 ± 0.33	18.22 ± 0.23	8.17 ± 0.04
	GG (*n* = 18)	38.38 ± 1.62	71.74 ± 1.76	67.55 ± 1.22	82.07 ± 1.33	32.87 ± 0.65	18.77 ± 0.77	17.41 ± 0.52	8.14 ± 0.10

BW = body weight; BL = body length; BH = body height; ChC = chest circumference; ChD = chest depth; ChW = chest width, HW = hip width; and CaC = cannon circumference. ^a,b^ Within a row, means with different superscript letters are significantly different (*p* < 0.05). ^A,B^ Within a row, means with different superscript letters are very significantly different (*p* < 0.01). ^1^ Data represent means ± SEM (*n* = 233).

**Table 6 genes-13-00077-t006:** Association analysis of haplotypes in *SSTR1* with growth traits of Hulun Buir sheep at birth and weaning ^1^.

Genotypes	BRW/kg	BW/kg	BL/cm	BH/cm	ChC/cm	ChD/cm	ChW/cm	HW/cm	CaC/cm
ACC (*n* = 176)	4.23 ± 0.05	23.13 ± 0.53	57.01 ± 0.57	56.04 ± 0.42	68.12 ± 0.56	27.97 ± 0.23	15.72 ± 0.16	12.49 ± 0.11	7.48 ± 0.04
ACG (*n* = 88)	4.25 ± 0.07	23.74 ± 0.75	57.93 ± 0.80	56.33 ± 0.60	68.88 ± 0.79	28.21 ± 0.33	15.86 ± 0.23	12.50 ± 0.16	7.50 ± 0.06
ATC (*n* = 25)	4.13 ± 0.14	20.48 ± 1.40	55.02 ± 1.50	54.30 ± 1.12	65.22 ± 1.49	27.74 ± 0.62	14.54 ± 0.77	12.06 ± 0.29	7.30 ± 0.11
ATG (*n* = 8)	4.30 ± 0.25	23.16 ± 2.47	58.81 ± 2.65	55.69 ± 1.98	67.19 ± 3.68	28.38 ± 1.10	14.94 ± 0.77	12.00 ± 0.51	7.50 ± 0.20
GCC (*n* = 36)	4.25 ± 0.12	23.17 ± 1.17	57.60 ± 1.25	56.39 ± 0.93	68.33 ± 1.24	28.46 ± 0.52	15.44 ± 0.36	12.26 ± 0.24	7.45 ± 0.10
GCG (*n* = 11)	4.22 ± 0.21	26.10 ± 2.11	60.96 ± 2.26	58.73 ± 1.69	72.00 ± 2.24	29.23 ± 0.94	16.36 ± 0.65	12.55 ± 0.44	7.62 ± 0.17
GTC (*n* = 3)	4.06 ± 0.40	17.40 ± 4.04	51.83 ± 4.33	52.33 ± 3.24	64.83 ± 4.29	27.17 ± 1.79	15.17 ± 1.25	12.50 ± 0.84	7.00 ± 0.33
*p*	0.987	0.223	0.258	0.357	0.211	0.805	0.123	0.753	0.512

BRW = birth weight; BW = body weight; BL = body length; BH = body height; ChC = chest circumference; ChD = chest depth; ChW = chest width, HW = hip width; CaC = cannon circumference. ^1^ Data represent means ± SEM (*n* = 233).

**Table 7 genes-13-00077-t007:** Association analysis of haplotypes in *SSTR1* with growth traits of Hulun Buir sheep at 9 months of age ^1^.

Genotypes	BW/kg	BL/cm	BH/cm	ChC/cm	ChD/cm	ChW/cm	HW/cm	CaC/cm
ACC (*n* = 176)	31.98 ± 0.57	66.58 ± 0.39	63.66 ± 0.33	82.90 ± 0.63	32.48 ± 0.23	21.45 ± 0.20	14.56 ± 0.12	7.53 ± 0.05
ACG (*n* = 88)	32.60 ± 0.81	67.10 ± 0.55	64.42 ± 0.47	84.07 ± 0.89	32.98 ± 0.32	22.10 ± 0.28	14.66 ± 0.16	7.56 ± 0.06
ATC (*n* = 25)	29.92 ± 1.51	64.54 ± 1.02	62.20 ± 0.88	80.88 ± 1.67	31.65 ± 0.59	20.96 ± 0.53	14.23 ± 0.31	7.48 ± 0.12
ATG (*n* = 8)	32.16 ± 2.67	66.63 ± 1.81	64.00 ± 1.56	83.94 ± 2.95	32.50 ± 1.07	21.13 ± 0.94	14.81 ± 0.55	7.25 ± 0.21
GCC (*n* = 36)	32.12 ± 1.26	67.57 ± 0.85	63.85 ± 0.74	84.40 ± 1.39	32.75 ± 0.51	22.25 ± 0.44	14.33 ± 0.26	7.67 ± 0.10
GCG (*n* = 11)	34.70 ± 2.28	69.86 ± 1.54	65.55 ± 1.33	87.96 ± 2.52	34.41 ± 0.91	22.82 ± 0.80	14.73 ± 0.47	7.64 ± 0.28
GTC (*n* = 3)	26.67 ± 4.36	64.00 ± 2.95	63.00 ± 2.55	76.33 ± 4.82	31.00 ± 1.75	22.00 ± 1.53	13.00 ± 0.90	7.50 ± 0.35
*p*	0.502	0.087	0.325	0.154	0.225	0.152	0.416	0.585

BW = body weight; BL = body length; BH = body height; ChC = chest circumference; ChD = chest depth; ChW = chest width, HW = hip width; CaC = cannon circumference. ^1^ Data represent means ± SEM (*n* = 234).

**Table 8 genes-13-00077-t008:** Association analysis of haplotypes in *SSTR1* with growth traits of Hulun Buir sheep at 16 months of age ^1^.

Genotypes	BW/kg	BL/cm	BH/cm	ChD/cm	HW/cm	CaC/cm
ACC (*n* = 176)	38.21 ± 0.46	72.44 ± 0.60	67.34 ± 0.34	33.97 ± 0.34	18.10 ± 0.16	8.23 ± 0.06
ACG (*n* = 88)	38.13 ± 0.64	73.24 ± 0.84	66.64 ± 0.49	33.53 ± 0.47	18.06 ± 0.22	8.16 ± 0.09
ATC (*n* = 25)	35.73 ± 1.19	69.19 ± 1.57	64.12 ± 0.90	32.92 ± 0.88	17.67 ± 0.41	8.14 ± 0.16
ATG (*n* = 8)	36.26 ± 2.15	70.38 ± 2.82	65.63 ± 1.63	33.38 ± 1.59	17.69 ± 0.73	8.13 ± 0.29
GCC (*n* = 36)	39.09 ± 1.01	72.01 ± 1.33	67.11 ± 0.77	35.25 ± 0.75	18.04 ± 0.35	8.38 ± 0.14
GCG (*n* = 11)	39.69 ± 1.84	75.09 ± 2.41	68.09 ± 1.39	34.18 ± 1.35	18.27 ± 0.63	8.18 ± 0.25
GTC (*n* = 3)	35.73 ± 3.51	74.00 ± 4.61	64.67 ± 2.65	33.33 ± 2.59	18.67 ± 1.20	8.00 ± 0.48
*p*	0.291	0.312	0.395	0.498	0.970	0.869

BW = body weight; BL = body length; BH = body height; ChD = chest depth; HW = hip width; and CaC = cannon circumference. ^1^ Data represent means ± SEM (*n* = 234).

## Data Availability

Not applicable.

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
