# Peer review of "Identification of Somatostatin Receptor Subtype 1 (SSTR1) Gene Polymorphism and Their Association with Growth Traits in Hulun Buir Sheep"

_genes, 2021, doi:10.3390/genes13010077_

Round 1
Reviewer 1 Report
In this study Authors amplified and sequenced thee exons of ovine SSTR1 gene in 233 Hulun Buir sheep to identify specific polymorphisms. They found 4 significant SNPs, 3 of them in strong linkage disequilibrium (SNP1, SNP3, and SNP4). Thereafore, their performed an association análisis between these SNPs and several growth traits previously recorded in the animals at birth, 4 months, 9months, and 16months of ages. Results showed that
SNP2 were associated with WW and BH at 4 months of age, and with BW, BL, ChC, HW traits at 9 months of age. SNP2 showed also a strong association with ChD at 4 and 9 months of age. SNP3 were associated with BH at 9 months of ages. SNP4 was associated with ChD, ChW, CaC at 9 months of age, while only with BW and ChC at 16 months of ages. Authors conclude that these results denostrated that SNP2, SNP3, and SNP4 could be use as molecular market for future selection for growth traits on Hulun Buir sheep.
General concept comments
I found the article interesting with a good experimental design. I claim a lack of good description of the methods such as the model used for association analysis. I suggest specifying better which are the fixed and random models, and how the combination factor between gender and genotype was handled
Moreover you should Include a table with simples information or specify including them in the text especially the gender information.
The new SNPs that you found have been deposited in Genbank or where did their ID come from? If yes you should specify it in the text.
Some more small suggestions are reported below:
- Line 18: type error “of”
- Line 13: type error “stateS”
- Line21: type error Hardy - Weinberg EquiLibrium
- Line 39: I suggest using a Single-nucleotide variant instead of
“The genetic variations”
- Line 69-73: I suggest to re-word in order to make it more fluent:
Despite the importance of SSTR1 in controlling the growth hormone axis, there is still a lack of researches on its effect on growth performance in sheep. Therefore, we conducted this study to explore the molecular characterization and polymorphisms in the exons of the SSTR1 gene and assess their association with several growth traits at different ages in Hulun Buir sheep.
- Line 74-78: you should delete these lines in this section, it is an introduction, not results or conclusion.
The manuscript is in general clear and relevant for the species described. Results are well described but I honestly think that more validation experiments will be necessary to affirm that the SNPs could be used as molecular markers for selection.
Author Response
Response to Reviewer 1 Comments
Dear Professors,
On behalf of all the contributing authors, I would like to express our sincere appreciations of your letter and reviewer’s constructive comments concerning our manuscript “Identification of somatostatin receptor subtype 1(SSTR1) gene polymorphism and their association with growth traits in Hulun Buir sheep” (Manuscript ID genes-1509833). These comments are all valuable and helpful for improving our article. According to your and reviewers’ comments, we have made modifications to our manuscript. In the revised version, changes to our manuscript were all highlighted within the documents by using red colored text. Point-by-point responses are listed below this letter.
General concept comments
1. I found the article interesting with a good experimental design. I claim a lack of good description of the methods such as the model used for association analysis. I suggest specifying better which are the fixed and random models, and how the combination factor between gender and genotype was handled.
Response: Thank you for your question. It was revised in the manuscript (Line 151-176).
2. Moreover you should Include a table with simples information or specify including them in the text especially the gender information.
Response: Thank you for your question. It was revised in the manuscript (Line 100).
3. The new SNPs that you found have been deposited in Genbank or where did their ID come from? If yes you should specify it in the text.
Response: Thank you for your question. These SNPs have been deposited in dbSNP database(As shown in figures). It was revised in the manuscript (Line 180). I’m sorry tI got the wrong number, they were revised in the manuscript (Line 181-183, 189,193,197,201,377).
Some more small suggestions are reported below:
1. Line 18: type error “of”
Response: Thank you for your question. It was revised in the manuscript (Line 25 ).
2. Line 13: type error “stateS”
Response: Thank you for your question. It was revised in the manuscript (Line 18).
3. Line21: type error Hardy - Weinberg EquiLibrium
Response: Thank you for your question. It was revised in the manuscript (Line 28 ).
4. Line 39: I suggest using a Single-nucleotide variant instead of“The genetic variations”
5. Response: Thank you for your question. It was revised in the manuscript (Line 52).
6. Line 69-73: I suggest to re-word in order to make it more fluent:
Despite the importance of SSTR1 in controlling the growth hormone axis, there is still a lack of researches on its effect on growth performance in sheep. Therefore, we conducted this study to explore the molecular characterization and polymorphisms in the exons of the SSTR1 gene and assess their association with several growth traits at different ages in Hulun Buir sheep.
Response: Thank you for your question. It was revised in the manuscript (Line 91-93).
7. Line 74-78: you should delete these lines in this section, it is an introduction, not results or conclusion.The manuscript is in general clear and relevant for the species described. Results are well described but I honestly think that more validation experiments will be necessary to affirm that the SNPs could be used as molecular markers for selection.
Response: Thank you for your question. It was revised in the manuscript (Line 94-96).

Reviewer 2 Report
The introduction should be rewritten, it seems unconnected.
Line 33-34: "Outstanding" should be replaced
Line 35: What do you mean by "stable heredity"?
Line 43: Please rephrase, english are not appropriate
Line 48: "is considered"
Line 59: "Evidence"
Lines 49-51 mention "SST and its receptors (SSTR) are widely distributed in the central nervous system, pancreas, intestines, stomach, kidney, liver, pancreas, lungs and placenta" and lines 62-63: ", there are five somatostatin receptor subtypes (SSTR1-5), which were exclusively highly expressed in brain". So the SSTRs are expressed exclusively in the brain or in these organs also?
Lines 49 and lines 64-65 are repeated information about GH secretion
Lines 74-76 are not introductory information, these are results
Line 97: 1.0 μL of DNA should be replaced with concentration units
Line 112: Reference 24 is not appropriate for HW testing. Probably a paper in the references of this study?
What kind of test was used to asseess HWE?
Table 2: Replace Gene Frequency with Allele Frequency
As I can see in table 2, the differences between Ho and He are quite large. Can you check again for HWE?
Lines 209-211 are irrelevant with this study
Lines 220-222: Please move these lines to Introduction
Lines 228-229: Rephrase
Lines 234-234: It's the third or fourth time the authors mention the functions of SSTRs and their relation with GH
The authors use "association" and "correlation" to describe the same thing, please be consistent
Lines 255-262: I think this topic should be discussed earlier, since it is not the main finding of the study. Also, the english need major corrections in this paragraph
Author Response
Response to Reviewer 2 Comments
Dear Professors,
On behalf of all the contributing authors, I would like to express our sincere appreciations of your letter and reviewer’s constructive comments concerning our manuscript “Identification of somatostatin receptor subtype 1(SSTR1) gene polymorphism and their association with growth traits in Hulun Buir sheep” (Manuscript ID genes-1509833). These comments are all valuable and helpful for improving our article. According to your and reviewers’ comments, we have made modifications to our manuscript. In the revised version, changes to our manuscript were all highlighted within the documents by using red colored text. Extensive editing of English language and style required in my manuscript, I used the English Editing Services of Genes, changes to our manuscript were all highlighted within the documents by using blue colored text. Point-by-point responses are listed below this letter.
- The introduction should be rewritten, it seems unconnected.
Response: Thank you for your question. It was revised in the manuscript (Line 44-96 ).
- Line 33-34: "Outstanding" should be replaced
Response: Thank you for your question. It was revised in the manuscript (Line 44 ).
- Line 35: What do you mean by "stable heredity"?
Response: Thank you for your question. It's a misrepresentation, was revised in the manuscript (Line 47).
- Line 43: Please rephrase, english are not appropriate
Response: Thank you for your question. It was revised in the manuscript (Line 43 ).
- Line 48: "is considered"
Response: Thank you for your question. It was revised in the manuscript (Line 58).
- Line 59: "Evidence"
Response: Thank you for your question. The sentence was rewritten in the manuscript (Line 66-68).
- Lines 49-51 mention "SST and its receptors (SSTR) are widely distributed in the central nervous system, pancreas, intestines, stomach, kidney, liver, pancreas, lungs and placenta" and lines 62-63: ", there are five somatostatin receptor subtypes (SSTR1-5), which were exclusively highly expressed in brain". So the SSTRs are expressed exclusively in the brain or in these organs also?
Response: Thank you for your question. It was revised in the manuscript (Line 60-63).
- Lines 49 and lines 64-65 are repeated information about GH secretion
Response: Thank you for your question. It was revised in the manuscript (Line 56-59).
- Lines 74-76 are not introductory information, these are results
Response: Thank you for your question. It was revised in the manuscript (Line 91-96).
- Line 97: 1.0 μL of DNA should be replaced with concentration units
Response: Thank you for your question. It was revised in the manuscript (Line 122).
- Line 112: Reference 24 is not appropriate for HW testing. Probably a paper in the references of this study?
Response: Thank you for your question. It was deleted (Line 142), and removed reference 24.
- What kind of test was used to asseess HWE?
Response: Thank you for your question. It was revised in the manuscript (Line 142).
- Table 2: Replace Gene Frequency with Allele Frequency
Response: Thank you for your question. It was revised in the manuscript (Table 2 ).
- As I can see in table 2, the differences between Ho and He are quite large. Can you check again for HWE?
Response: Thank you for your question. In the previous manuscript, Ho stood for homozygosity, so the differences between Ho and He are quite large. In this modification, Ho was revised to observe heterozygosity, Ho, He, PIC, Na and HWE were checked and recalculated, It was revised in the manuscript (Table 2 ).
- Lines 209-211 are irrelevant with this study
Response: Thank you for your question. But I am sorry that did not find the sentence needed to be modified.
- Lines 220-222: Please move these lines to Introduction
Response: Thank you for your question. It was revised in the manuscript (Line 78-81).
- Lines 228-229: Rephrase
Response: Thank you for your question. It was revised in the manuscript (Line 342).
- Lines 234-234: It's the third or fourth time the authors mention the functions of SSTRs and their relation with GH
Response: Thank you for your question. It was revised in the manuscript (Line 375).
- The authors use "association" and "correlation" to describe the same thing, please be consistent
Response: Thank you for your question. It was revised in the manuscript (Line 23, 304, 322, 335).
- Lines 255-262: I think this topic should be discussed earlier, since it is not the main finding of the study. Also, the english need major corrections in this paragraph
Response: Thank you for your question. It was revised in the manuscript (Line 363-369).

Round 2
Reviewer 2 Report
Thank you for addressing all my questions as well as the questions and comments of reviewer 1.